# Pulmonary Edema Severity Estimation in Chest Radiographs Using Deep Learning

**Xin Wang[1], Evan Schwab[1], Jonathan Rubin[1], Prescott Klassen[1], Ruizhi Liao[2], Seth Berkowitz[3], Polina Golland[2], Steven Horng[3], Sandeep Dalal[1]**

[1] *Philips Research North America, Cambridge, MA, USA*
[2] *Computer Science and Artificial Intelligence Lab, MIT, Cambridge, MA, USA*
[3] *Beth Israel Deaconess Medical Center, Boston, MA, USA*

## Abstract

The detection of pulmonary edema in chest radiographs is critical for the physician to make timely treatment decisions for patients with congestive heart failure. However, assessing the severity of pulmonary edema is a challenging task that leads to low inter-rater agreement among experienced radiologists. We compare a number of deep learning approaches to estimate the severity of pulmonary edema using the large-scale MIMIC-CXR database of chest x-ray images and radiology reports.

**Keywords:** congestive heart failure, pulmonary edema severity, chest x-ray, deep learning

## 1. Introduction

Pulmonary edema is one of the most direct symptoms of congestive heart failure (CHF) and it shows up in chest radiographs as opacities in the lungs, thickening of the bronchial walls, increased interstitial markings, and hazy contours of blood vessels. However, accurate assessment of the severity of pulmonary edema is challenging because it relies on much more subtle findings (Halperin et al., 1985) (see Fig. 1). Experienced radiologists and ED physicians demonstrated a sensitivity of only 77% and 59% respectively when detecting the presence of edema, and their inter-rater agreement is low (Hammon et al., 2014; Kennedy et al., 2011). Deep learning has been used in previous studies to detect the presence of edema in radiographs (Rajpurkar et al., 2017; Wang et al., 2018; Rubin et al., 2018) but not to estimate severity.

In this work, we compare multiple supervised and semi-supervised deep learning methods to predict the severity of edema from radiograph images using the large-scale clinical MIMIC-CXR database (Johnson et al., 2019) consisting of 473,064 chest x-ray (CXR) images and 206,574 radiology reports collected from 63,478 patients in the ED and subsequent in-hospital stay. Of this total, a subset of patients have been diagnosed as having active CHF during their ED visit. We extract the edema severity labels from the radiology reports associated with CHF patients by searching for keywords that are highly correlated with stages of edema yielding 5,771 frontal CXR images which we confidently labeled as one of four severity levels: no edema (1,003), mild edema (3,058), moderate edema (1,414), and severe edema (296). We model this problem as a multi-class classification task to predict the level of severity of pulmonary edema.

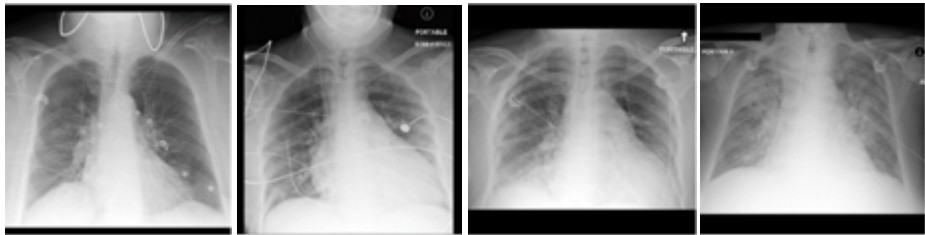

Figure 1: Example MIMIC-CXR images from the CHF cohort with (from left to right) no edema, mild edema, moderate edema, and severe edema. The differences in severity is subtle and challenging to diagnosis consistently by radiologists.

## 2. Methods

We randomly split the 5,771 labeled images into training (80%), validation (10%) and test set (10%) with no patient overlap and the same class distribution across sets. For evaluation, we use multi-class AUC given that accuracy has significant limitations in the context of imbalanced datasets. Multi-class AUC is calculated by taking the average of AUCs obtained independently for each class for the binary classification task of distinguishing a given class from all the other classes.

**Supervised Learning** First, we used a DenseNet (Huang et al., 2017) architecture and compared performance by training from scratch vs. with weights initialized from pre-training on ImageNet. We found that pre-training with ImageNet was superior. Then, we investigated three approaches to mitigate class imbalance: 1) weighted cross-entropy loss, 2) class-aware sampling (Shen et al., 2016), and 3) random minority oversampling (RMO) (Buda et al., 2018) with RMO outperforming the others. Using RMO and pre-trained ImageNet weights, we compared DenseNet against 4 other models: ResNet50 (He et al., 2016), InceptionV3, InceptionResNetV2 and NASNetMobile (Zoph et al., 2018). We apply the Adam optimizer with scheduled reduced learning rate, a mini-batch size of four, and online data augmentation. We downscale the images to $512 \times 512$ while keeping the original image aspect ratio by zero padding.

In addition, we investigated the effect of focusing on the lung region of interest (ROI) by cropping the lung ROIs from the original CXR. The detection of the lung is based on its segmentation using a fully convolutional DenseNet (Jégou et al., 2017) trained on two publicly available CXR data sets (Jaeger et al., 2014; Shiraishi et al., 2000).

**Semi-Supervised Learning** To improve classification performance, we facilitate utilization of the large number of unlabeled images in the MIMIC-CXR via semi-supervised learning based on self-training with pseudo labeling (Lee, 2013; Li et al., 2018; Wu and Prasad, 2018). We generate pseudo labels for unlabeled images in the CHF cohort using the DenseNet model we trained with RMO and pre-training, and enlarge the original training set by including the pseudo labeled data. We compared two different methods for including pseudo labels in the training set.

For method I, we take the union of the labeled images and all 15,777 pseudo labeled images. During training, we use a weighted loss function to mitigate the imbalance between the number of images with true labels and pseudo labels. We also apply a scheduled slowly increasing coefficient (Lee, 2013) to the loss term for the pseudo labels to ensure training with the labeled data does not get disturbed and benefits from unlabeled data while avoiding poor local minima (Grandvalet and Bengio, 2006).

For method II, we select the same number of pseudo labeled images as true labels (4,537) with the criteria of high prediction confidence. Instead of using prediction probability as the indicator of confidence, we apply the Monte-Carlo Dropout method (Gal and Ghahramani, 2016) to estimate a distribution for each prediction by repeating prediction on the unlabeled images 1,000 times with random dropout. Then, to get a better estimate of prediction confidence we subtract the standard deviation from the mean of the prediction probability.

Table 1: Comparison of multi-class AUC on the test set using DenseNet, ResNet50, InceptionV3, InceptionResNetV2, NASNetMobile, DenseNet w/ Lung ROI, and DenseNet w/ Semi-Supervised Model I and II.

| Severity | Dense | Res | Incept | InRes | NAS | Lung ROI | Semi-I | Semi-II |
|----------|-------|-----|--------|-------|-----|----------|--------|---------|
| **No** | 0.837 | 0.772 | 0.806 | 0.829 | 0.822 | 0.804 | 0.839 | **0.853** |
| **Mild** | 0.710 | 0.675 | 0.658 | 0.710 | 0.714 | 0.717 | 0.724 | **0.747** |
| **Moderate** | 0.755 | 0.726 | 0.697 | 0.748 | **0.788** | 0.770 | 0.774 | 0.772 |
| **Severe** | 0.843 | 0.826 | 0.833 | 0.814 | 0.840 | 0.885 | **0.889** | 0.879 |
| **Multi-Class** | 0.786 | 0.750 | 0.749 | 0.774 | 0.791 | 0.794 | 0.807 | **0.813** |

## 3. Results

The test AUC of each method is presented in Table 1. Among the five different supervised learning methods, NASNetMobile outperformed the others with multi-class AUC of 0.791. The DenseNet model trained with the cropped lungs achieved higher AUC for all severity levels except for No edema compared with DenseNet applied on the whole image. Finally, both the semi-supervised models trained with pseudo labels improved the classification over the supervised learning. The model trained with all pseudo labels (method I) performs slightly worse than the model trained with only a small subset of pseudo labels with high confidence. This is probably due the errors of the pseudo-labels with low confidence are reinforced during the self-training.

## 4. Conclusion

We demonstrated that deep learning is a promising approach for estimating pulmonary edema from chest radiographs. In addition, by using semi-supervised learning via self training with pseudo labeling we are able to make use of the large-scale unlabeled images to improve classification performance.

## Acknowledgments

This work was partially supported by MIT-Philips Collaboration grant, Wistron, and AWS. The authors gratefully acknowledge Alistair Johnson from the MIT Laboratory for Computational Physiology for curating and making the MIMIC-CXR dataset available.

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
