# OpenReview forum: "Pulmonary Edema Severity Estimation in Chest Radiographs Using Deep Learning"
_MIDL.io/2019/Conference/Abstract — MIDL Abstract 2019_

### Official Review · AnonReviewer2 · 2019-04-30
**Comparison of different neural networks and semi-supervised learning techniques for edema severity estimation.**

**Rating:** 3
**Confidence:** 3

**Review:**

This paper provides a comparison study of 5 neural network architectures, 3 class imbalance mitigation strategies and 2 semi-supervised learning techniques to achieve the best possible score on a edema severity classification task. This work work may offer some direction to the community when optimizing neural network hyperparameters for other tasks.

How significant are the results and do the authors believe that the ordering of the performance may be different for a rerun of the experiments or in a cross validation setting?

Did the authors consider data augmentation? On certain problems simple data augmentation strategies have been shown to perform just as well as semi supervised learning. (see Oliver et al. "Realistic Evaluation of Deep Semi-Supervised Learning Algorithms")

---

### Official Review · AnonReviewer1 · 2019-05-02
**Benchmark of DL models for severity estimation**

**Rating:** 2
**Confidence:** 2

**Review:**

The proposed work tests and benchmarks several architectures and methodologies for estimation of edema severity. It explores pretrained vs non-pretrained networks, different models for correcting for label unbalance, and multiple architectures such as Resnet, Inception and NAS. It also shows the effect of joint supervised+unsupervised training to increase the sample size and variability. Results show that the NAS model has the highest mean accuracy (point estimate) among supervised methods, while semi-II (Monte-Carlo Dropout) had the highest accuracy (point estimate) overall. While the paper is well executed, it does note really provide new insights towards solving the problem beyond a benchmark table with point estimates. For example, why is the problem of severity estimation technically different from edema classification? why would the optimal solution be different? Is the semi-II really better than other methods (no statistical tests of significant improvement)? or was this just a lucky outcome given this specific data partition?  If a paper only proposes a benchmark, the statistical comparison has to be sound and not just based on point estimates of accuracy.

---

### Decision · Program_Chairs · 2019-05-06
**Acceptance Decision**

Accept